# Efficient estimation of neural tuning during naturalistic behavior

**Edoardo Balzani**
Center for Neural Science
New York University
New York, NY, 10003
eb162@nyu.edu

**Kaushik Lakshminarasimhan**
Center for Theoretical Neuroscience
Columbia University
New York, NY, 10027
jl5649@columbia.edu

**Dora E. Angelaki**
Center for Neural Science
New York University
New York, NY, 10003
da93@nyu.edu

**Cristina Savin**
Center for Neural Science
Center for Data Science
New York University
New York, NY, 10003
cs5360@nyu.edu

## Abstract

Recent technological advances in systems neuroscience have led to a shift away from using simple tasks with low-dimensional, well-controlled stimuli towards trying to understand neural activity during naturalistic behavior. However, with the increase in number and complexity of task-relevant features, standard analyses such as estimating tuning functions become challenging. Here, we use a Poisson generalized additive model (P-GAM) with spline nonlinearities and an exponential link function to map a large number of task variables (input stimuli, behavioral outputs, and activity of other neurons, modeled as discrete events or continuous variables) into spike counts. We develop efficient procedures for parameter learning by optimizing a generalized cross-validation score and infer marginal confidence bounds for the contribution of each feature to neural responses. This allows us to robustly identify a minimal set of task features that each neuron is responsive to, circumventing computationally demanding model comparison. We show that our estimation procedure outperforms traditional regularized GLMs in terms of both fit quality and computing time. When applied to neural recordings from monkeys performing a virtual reality spatial navigation task, P-GAM reveals mixed selectivity and preferential coupling between neurons with similar tuning.

## 1 Introduction

From rodent decision-making in dynamic environments [1, 2] to complex virtual reality setups for freely behaving animals [3, 4], there is a growing interest in studying neural activity in the context of naturalistic behavior [5]. This not only increases the number and complexity of the task-relevant variables, but also sacrifices precise experimental control of their statistical properties, making even the most basic analyses such as estimating tuning functions increasingly challenging.

During complex behavior, it is often unclear which features the neurons may be tuned to, especially in higher cortical regions. There is a natural temptation to use very flexible models that consider all possible task features and let the estimation procedure determine which input dimensions actually drive the neural responses. While regularized GLMs are the model of choice for this estimation, they require careful design of basis functions [6, 7]; they are also nontrivial to regularize well [8] or to

scale to large datasets [9]. Worse still, deciding which input dimensions should be included requires costly model comparison, making it unfeasible for high dimensional inputs. More sophisticated alternatives, based on Gaussian Process (GP) priors [10, 11, 12, 13] are most flexible, but scale unfavorably with respect to both the number of input dimensions and data size [14, 12]. Overall, it remains unclear how to model complex neural responses in a robust and scalable manner.

Here we use computationally efficient low-rank Generalized Additive Models (GAM) based on penalized regression splines to overcome the statistical challenges associated with modeling neural responses in the context of naturalistic behavior. At the core of our solution is smoothness-enforcing regularization, paired with efficient optimization based on an iterative procedure that jointly optimizes both parameters and hyperparameters.[1] Given the (hyper)parameter estimates, we reinterpret the penalties as a prior to derive a posterior for the parameters, which in turn is used to individually assess if each input feature has a statistically significant contribution to neural activity and should be included in the model. We demonstrate the efficiency of this procedure using artificial data, showing that P-GAM outperforms standard GLMs. When applied to neural recordings from monkeys performing a spatial navigation task in virtual reality, P-GAM recovers known features of the neural code, in particular mixed selectivity[15] and structured noise correlations [16, 17, 18].

## 2   Coupled Poisson Generalized Additive Model

We model spike counts $\boldsymbol{y}_t \in \mathbb{Z}_{\geq 0}^N$ of a neural population as a nonlinear function of continuous covariates $\boldsymbol{x}(t) \in \mathbb{R}^K$, binary events $\boldsymbol{z}(t) \in \{0,1\}^H$, and past neural activity $\boldsymbol{y}_{1:t-1}$ (Fig.1A). Formally, each neuron's responses are modelled as a Poisson GAM [19] with sufficient statistics given as a sum of nonlinear functions of the covariates and an exponential link function:

$$\log(\mu_t^i) = \sum_{j=1}^{K} f_j^i(x_j(t)) + \sum_{j=1}^{H} k_j^i * z_j(t) + \sum_{j=1}^{N} y^{(j)} * h_j^i(t) + c \tag{1}$$

$$y_t^{(i)}|\boldsymbol{y}_{1:t-1}, \boldsymbol{x}, \boldsymbol{z}, \boldsymbol{f}, \boldsymbol{k}, \boldsymbol{h}, c \sim \text{Poisson}\left(\mu_t^i\right), \tag{2}$$

where $f_j^i(\cdot)$ are smooth functions of individual input features, $k_j^i(\cdot)$ are smooth temporal kernels describing the response to task events, $h_j^i(\cdot)$ are smooth causal filters (e.g. $h_j^i(t) = 0$ if $t < 0$) capturing the directional coupling from neuron $j$ to neuron $i$, with an auto-regressive component for $i = j$, which accounts for refractory period effects; $*$ is the convolution operator, and $c$ is a constant capturing baseline firing. The same formalism can be readily extended to spatio-temporal nonlinear filters or other two-dimensional dependencies (see Suppl. Info. S7 and Fig. S2, S3), although enforcing smoothness across many inputs is difficult.

The factorization of this conditional makes the joint likelihood relatively simple:

$$\begin{aligned} l_L &= \log p(y_{1:T}^{(i)}|\boldsymbol{y}_{1:T}^{[-i]}, \boldsymbol{x}, \boldsymbol{z}, \boldsymbol{f}, \boldsymbol{k}, \boldsymbol{h}, \boldsymbol{c}) \\ &= \sum_{t=1}^{T} \log p(y_t^{(i)}|\boldsymbol{y}_{1:t-1}, \boldsymbol{x}, \boldsymbol{z}, \boldsymbol{f}, \boldsymbol{k}, \boldsymbol{h}, \boldsymbol{c}) = \sum_{t=1}^{T} \log p(y_t^{(i)}|\mu_t), \end{aligned} \tag{3}$$

where $\boldsymbol{y}_{1:T}^{[-i]}$ denotes the spike counts of all neurons except the $i^{\text{th}}$.

**Prior for smooth functions.** We need to define priors over nonlinear functions that encourage smoothness, while keeping computations tractable. To this end, we model functions $f_j, k_j, h_j$ using a penalized spline basis expansion. We use a finite and local basis set of splines of order $m$, interpolating between a fixed set of knots (see S1 for details) [20]. The order of the splines and the knots locations are model hyperparameters that could be optimized by cross-validation; in practice, we use cubic splines ($m = 4$) and manually choose knots that reasonably cover the input range.

Having specified the basis set $\boldsymbol{b}$, we can expand each response function $f(\cdot)$ as[2]

$$f(x) = \sum_{j=1}^{m} b_j(x)\beta_j = \boldsymbol{b}(x)^{\top}\boldsymbol{\beta},$$

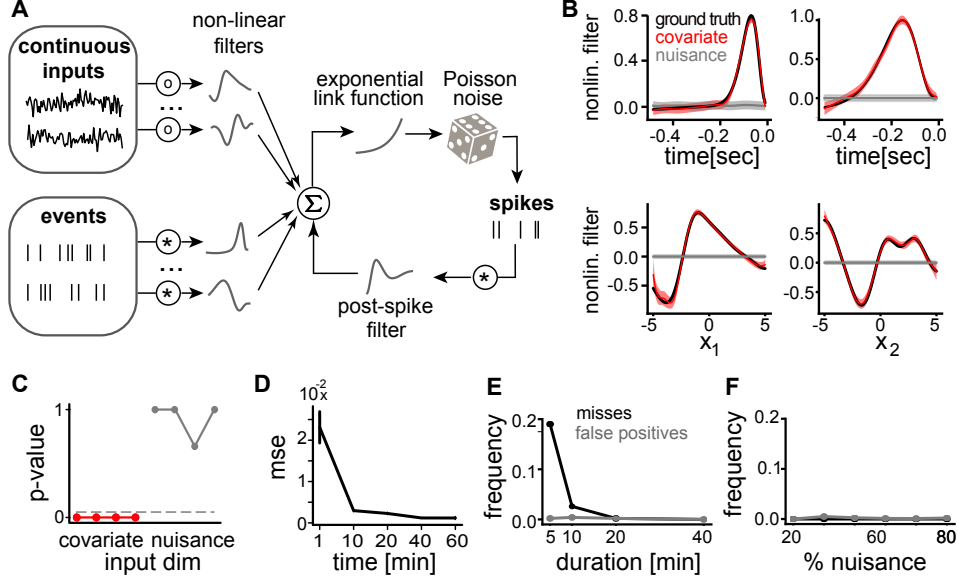

Figure 1: **A**. P-GAM generative model: spikes are modeled as Poisson counts with a mean given as the sum of a set of smooth nonlinear response functions. **B**. Example filter estimates for two binary (top) and two continuous (bottom) covariates; for each input we include an additional nuisance dimension, which is correlated with the true covariate but does not drive neural responses. Ground truth in black, estimates for true covariates and nuisance variables in red and gray, respectively; shaded areas show 99% confidence intervals. **C**. P-values for both true covariates (red) and nuisance inputs (gray), significance threshold p=0.01 as dashed line. **D**. For the same setup, mean squared error for log mean firing rates as a function of the amount of data available. **E**. Model selection: false positive and false negative rates in input variable selection as a function of simulation duration and **F**. fraction of nuisance inputs. Error bars show $\pm 1$ s.e.m. estimated using 100 independent repeats.

where $m$ is the number of basis functions. To enforce a smooth prior, each spline is associated with a quadratic penalty term that controls its energy ('wiggliness'):

$$\mathcal{L}_j(\lambda) = \lambda \int f_j''(x)^2 dx = \boldsymbol{\beta}^\top \boldsymbol{S}_{f_j} \boldsymbol{\beta},$$

with penalty matrix $\boldsymbol{S}_{f_j} = \lambda \int \boldsymbol{b}'' \boldsymbol{b}''^\top dx$ and a smoothing penalty $\lambda \in \mathbb{R}^+$ that individually controls the degree of smoothness for each input feature. The role of $\lambda$ is to limit model degrees of freedom, with larger $\lambda$s resulting in increasingly smooth models.

Pooling together the penalties for each term, we get the full smoothing regularizer as[3]

$$\mathcal{L}(\boldsymbol{\lambda}) = \sum_j \lambda_{f_j} \boldsymbol{\beta}_j^\top \boldsymbol{S}_{f_j} \boldsymbol{\beta}_j + \sum_j \lambda_{k_j} \boldsymbol{\alpha}_j^\top \boldsymbol{S}_{k_j} \boldsymbol{\alpha}_j + \sum_j \lambda_{h_j} \boldsymbol{\delta}_j^\top \boldsymbol{S}_{h_j} \boldsymbol{\delta}_j = \boldsymbol{\beta}^\top \boldsymbol{S}_{\boldsymbol{\lambda}} \boldsymbol{\beta}, \qquad (4)$$

where $f_j = \boldsymbol{b}_j(x)^\top \boldsymbol{\beta_j}$, $k_j = \boldsymbol{a}_j(t)^\top \boldsymbol{\alpha_j}$, $h_j = \boldsymbol{d}_j(t)^\top \boldsymbol{\delta_j}$, we are stacking all the regression coefficient in a single column vector, $\boldsymbol{\beta} = [c; \boldsymbol{\beta_1}; \dots \boldsymbol{\beta}_K; \dots; \boldsymbol{\alpha}_1; \dots; \boldsymbol{\alpha}_H; \boldsymbol{\delta}_1; \dots; \boldsymbol{\delta}_N]$, and all the penalties in a single block-diagonal matrix, $\boldsymbol{S}_{\boldsymbol{\lambda}}$.

Putting everything together, the penalized log-likelihood becomes

$$l_p(\boldsymbol{\beta}|\boldsymbol{\lambda}) = l_L(\boldsymbol{\beta}) - \frac{1}{2} \boldsymbol{\beta}^\top \boldsymbol{S}_{\boldsymbol{\lambda}} \boldsymbol{\beta}, \qquad (5)$$

where $l_L(\boldsymbol{\beta})$ is the log-likelihood in (3).

# 3 Parameter estimation

The spline basis expansion transforms the GAM into an over-parametrized GLM, with parameters $\boldsymbol{\alpha}_j$, $\boldsymbol{\beta}_j$, $\boldsymbol{\delta}_j$ and $\boldsymbol{\lambda}$. For any fixed $\boldsymbol{\lambda}$, the regression coefficients can be learned by maximizing the penalized log-likelihood (5), which we do here via the penalized iteratively re-weighted least-squares algorithm (PIRLS) [21]. To optimize $\boldsymbol{\lambda}$, we use a modified version of the GCV, the Double Generalized Cross Validation score (dGCV, [22]), which aims to maximize prediction accuracy while controlling for over-fitting. The joint parameter optimization is done by a modified version of performance oriented iteration [23], which alternates between optimizing the penalties and updating the regression weights. Moreover, thanks to the factorization assumptions of the penalized log likelihood, this optimization can be performed in parallel for each unit.

**Penalized iterative re-weighted least squares.** Before describing the PIRLS for P-GAM, we again refer to our optimization objective,

$$\underset{\boldsymbol{\beta}}{\operatorname{argmax}}\, l_p(\beta|\lambda) = \underset{\boldsymbol{\beta}}{\operatorname{argmax}}\, \sum_t \log p(y_t|\mu_t) - \frac{1}{2}\boldsymbol{\beta}^\top \boldsymbol{S_\lambda}\boldsymbol{\beta}, \tag{6}$$

where $p(y_t|\mu_t)$ is the likelihood of a Poisson variable with mean $\mu_t = \exp(\boldsymbol{X}_t\boldsymbol{\beta})$; for notational convenience, we have pooled all basis functions evaluated at time $t$ in a model matrix $\boldsymbol{\beta}$ and all the inputs in a design matrix, $\boldsymbol{X}_t = [\mathbf{1} : \boldsymbol{b}_1 \cdots \boldsymbol{b}_K : \boldsymbol{a}_1 * z_1 \cdots \boldsymbol{a}_H * z_H : \boldsymbol{d}_1 * y^{(1)} \cdots \boldsymbol{d}_N * y^{(N)}]$, where : marks horizontal concatenation.

The Newton optimization for this loss takes the form of the following weighted least squares loop:

1. initialize $\hat{\mu}_t = y_t + \delta_t$, where $\delta_t$ usually set to 0 or to a small positive constant. $\hat{\eta}_t = \log(\hat{\mu}_t)$
2. compute the pseudo-data $z_t = \frac{1}{\hat{\mu}_t}(y_t - \hat{\mu}_t) + \hat{\eta}_t$ and the iterative weights $w_t = \hat{\mu}_t$
3. find $\hat{\boldsymbol{\beta}} = \underset{\boldsymbol{\beta}}{\operatorname{argmin}}\, \|z - X\beta\|^2_{\boldsymbol{W}} + \boldsymbol{\beta}^\top \boldsymbol{S_\lambda}\boldsymbol{\beta}$, with $\boldsymbol{W}$ the diagonal matrix with $\boldsymbol{W}_{tt} = w_t$
4. update $\hat{\eta}_t = \boldsymbol{X}_t\hat{\boldsymbol{\beta}}$, $\hat{\mu}_t = e^{\hat{\eta}_t}$
5. repeat 2-4 until convergence

**Double Generalized Cross Validation score.** The role of dGCV score optimization is to learn the appropriate degree of smoothness for the non-linearities from the data. To define it, it is useful to start from PIRLS, rewriting the least squares component (step 3) as

$$\|z - X\beta\|^2_{\boldsymbol{W}} + \boldsymbol{\beta}^\top \boldsymbol{S_\lambda}\boldsymbol{\beta} = \|\sqrt{\boldsymbol{W}}z - \boldsymbol{A}(\boldsymbol{\lambda})\sqrt{\boldsymbol{W}}z\|^2 + \boldsymbol{\beta}^\top \boldsymbol{S_\lambda}\boldsymbol{\beta},$$

where $\boldsymbol{A} = \sqrt{\boldsymbol{W}}\boldsymbol{X}(\boldsymbol{X}^\top\boldsymbol{W}\boldsymbol{X} + \boldsymbol{S_\lambda})^{-1}\boldsymbol{X}^\top\sqrt{\boldsymbol{W}}$ is sometimes referred to as the influence matrix and $z$ is the pseudo data computed in PIRLS step 2. A simple cross-validation procedure is to predict the observation at one specific time point, conditioned on observations at all other time points, where we define $\hat{y}^{[-j]}$ as the predicted $y$ conditioned on all time points with the exception of $t = j$. With a little algebra, we can show that the leave-one-out prediction error can be expressed as $y_t - \hat{y}_t^{[-t]} = (y_t - \hat{y}_t)/(1 - A_{tt})$, where $\hat{y}_t$ is the prediction conditioned on the full data, and $A_{tt}$ is the diagonal entry of the influence matrix [24]. This allows us to compute the leave-one-out Ordinary Cross Validation (OCV) for a single model fit on the complete dataset as

$$\text{OCV}(\boldsymbol{\lambda}) = \frac{1}{n}\sum_t (y_t - \hat{y}_t^{[-t]})^2 = \sum_t \frac{(y_t - \boldsymbol{A}_{t:}\boldsymbol{y})^2}{n(1 - A_{tt})^2}.$$

This is an intuitive metric, but not without problems. In particular, OCV weights differently the reconstruction error for different time points, and, unlike the original least-square score, it is not rotation invariant. These problems can be corrected by rewriting the score as a rotated LS problem, which leads to the GCV score [24], with a further modification shown to reduce the risk of overfitting [22]. This leads to our final dGCV:

$$\text{dGCV}(\boldsymbol{\lambda}) = \frac{n\|\boldsymbol{y} - \boldsymbol{A}\boldsymbol{y}\|^2}{(n - \gamma\text{tr}(\boldsymbol{A}))^2}. \tag{7}$$

Here $\gamma$ is a constant greater than one, the larger the $\gamma$ the smoother the solution ($\gamma = 1.5$ in our simulations). To get some intuition for the role of $\gamma$, we can note that the trace of $\boldsymbol{A}$ represents the

model's effective degrees of freedom, as explained in Suppl. Info. S2. Therefore, we can interpret the dGCV as the mean squared prediction error, normalized by $\left(1 - \gamma \frac{\mathrm{tr}(\boldsymbol{A})}{n}\right)^2$, a factor which will penalize models with many degrees of freedom. The larger the $\gamma$, the stronger the penalization, leading to simpler (smoother) functional dependencies. Our choice of score is motivated by previous theoretical work showing that that dGCV combines the GCV score with a measure of prediction stability $c = \sum_t \left(\hat{y}_t - \hat{y}_t^{[-t]}\right)^2$, see [21]. For completeness, we include gradients and Hessian for dGCV in the Suppl. Info. S3.

**Performance oriented iteration.** Jointly optimizing $\beta$ and $\lambda$ parameters works as follows:

1. initialize $\boldsymbol{\lambda}$, usually setting all penalties to a small positive value
2. initialize the $\boldsymbol{\beta}$ as for PIRLS
3. perform one iteration of PIRLS to update $\boldsymbol{\beta}$
4. update the penalties by setting $\hat{\boldsymbol{\lambda}} = \underset{\lambda}{\mathrm{argmin}}\ \mathrm{dGCV}(\boldsymbol{\lambda})$ using e.g. conjugate gradients
5. repeat until convergence

Convergence can be defined in terms of both the dGCV score or the penalized log-likelihood. Empirically, we found that convergence speed can be improved by first performing a full optimization of $\boldsymbol{\beta}$ for fixed penalties, followed by the procedure described above.

# 4 Model selection

**Confidence intervals for parameters.** Beyond the flexible specification of the response function, one major advantage of GAMs with automatic smoothness learning is the availability of reliable confidence intervals (CI). In particular, Woods at al. [25] provide an analytic form for the asymptotic posterior probability of the regression coefficients: $\boldsymbol{\beta}|\boldsymbol{y}, \boldsymbol{X}, \boldsymbol{\lambda} \sim \mathcal{N}(\hat{\beta}, \boldsymbol{V}_\beta)$, where $\hat{\beta} = (\boldsymbol{X}^\top \boldsymbol{W} \boldsymbol{X} + \boldsymbol{S}_{\boldsymbol{\lambda}})^{-1} \boldsymbol{X}^\top \boldsymbol{W} \boldsymbol{z}$ is the maximizer of the penalized loss (Eq. 5), and $\boldsymbol{V}_\beta = (\boldsymbol{X}^\top \boldsymbol{W} \boldsymbol{X} + \boldsymbol{S}_{\boldsymbol{\lambda}})^{-1}$ is the posterior covariance, with $\boldsymbol{W}$ the diagonal matrix with entries $w_{tt} = e^{X_t \hat{\beta}}$.

The Gaussian approximation together with the spline basis means that uncertainty about individual $\beta$ elements can be easily translated into uncertainty about the corresponding nonlinear kernels and the final log firing rate, by marginalizing out all the $\beta$ elements corresponding to the relevant single input feature(s).[4] The posterior mean for a single nonlinear response function $f(x)$ is $\hat{\boldsymbol{f}} = \tilde{\boldsymbol{X}}\hat{\boldsymbol{\beta}}$, where $\tilde{\boldsymbol{X}}$ is the model matrix with zeros in the columns that are not related to $\hat{f}(x)$. Since these marginals are also Gaussian, $\hat{f}_t \pm z_{\alpha/2}\sqrt{v_t}$ are approximate $100(1-\alpha)\%$ Bayesian credible confidence intervals for $v_t = \mathrm{diag}(\tilde{\boldsymbol{X}} \boldsymbol{V}_\beta \tilde{\boldsymbol{X}}^\top)$ and $z_{\alpha/2}$ is the $\alpha/2$ quantile of the standard normal distribution. Marra et al. [26] showed that for GAM with automatic smoothing, the Bayesian credible intervals also satisfy the frequentist coverage (Average Coverage Probability):

$$\mathrm{ACP} = \frac{1}{T}\sum_{t=1}^{T} \mathbb{P}\left(|f(x_t) - \hat{f}(x_t)| \leq z_{\alpha/2}\sqrt{v_t}\right) < 1 - \alpha, \tag{8}$$

where $f(x)$ is the true response function. We sketch the proof in (S4), emphasizing the points where the penalty learning is relevant for the good coverage properties of the confidence intervals.

**Additional regularization encouraging minimal models.** The neuroscientific question of interest for model comparison is which input features actually affect neural responses and thus should be included in a minimal model of the data. In GAM terms, this translates into selecting the subset of response functions that statistically significantly affect the firing rate of a unit.

Smoothing penalty learning already takes care of a large part of the model selection by assigning large penalties to responses that do not increase the model's predictive power. As penalty $\lambda_f$ in (4) grows, the corresponding maximum likelihood regression weights will be forced to live in the null space of $\boldsymbol{S}_{\lambda_f}$, which is the space of straight lines ($f''(x) = 0$). This means that maximally penalized

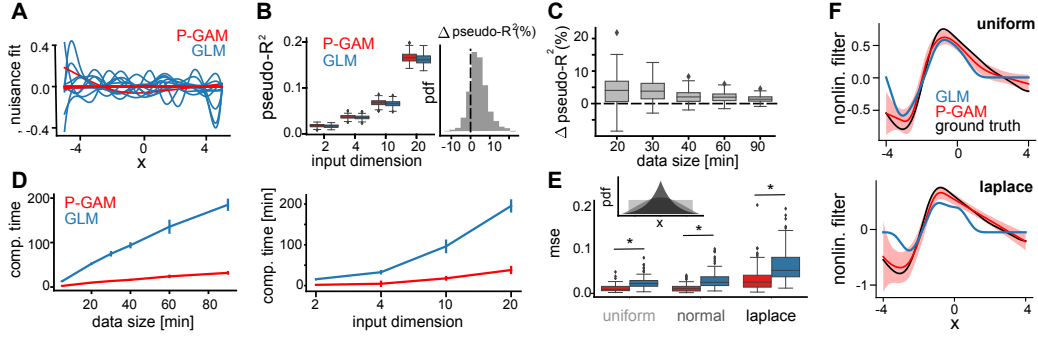

Figure 2: Comparison with traditional regularized GLM regression. **A**. Examples of residual nuisance response functions estimated by P-GAM (red) or by a GLM regression, with L1 and L2 regularization (blue). None of the P-GAM fits past the significance test. **B**. Cross-validated pseudo-$R^2$ estimates comparison; right: fractional improvements of P-GAM relative to GLM. **C**. P-GAM vs. GLM fit quality improvement as a function of dataset size, $T$. **D**. Computational time required for a full fit, as a function of data size (left) and input dimensionality (right). **E**. Different priors for input statistics (inset) and corresponding mean squared error between true and reconstructed responses (in parameter space). **F**. Example estimated filters for two input distributions.

response functions will not be completely zeroed out, just forced to be linear. Ideally, we would like the penalty matrix to be full rank so that large penalties would result in null responses; in this scenario, learning smoothing penalties will automatically remove unnecessary covariates. This can be achieved by adding an additional penalization term to (4), acting on the null space of $S_\lambda$ [21]. To do so, we perform an eigen-decomposition $S_\lambda = \boldsymbol{U}^\top \boldsymbol{DU}$, and we set $\tilde{\boldsymbol{U}}$ to be the matrix whose columns are the 0-eigenvectors of $S_\lambda$. The new penalty takes the form:

$$\mathcal{L}(\lambda, \lambda') = \boldsymbol{\beta}^\top (\boldsymbol{S}_\lambda + \lambda' \tilde{\boldsymbol{U}}^\top \tilde{\boldsymbol{U}}) \boldsymbol{\beta}. \tag{9}$$

The two penalties acts independently on orthogonal subspaces of the response function space, finally resulting in a penalization matrix with the desired zeros-only null space.

**Statistical significance for individual input dimensions.** We take advantage of the availability of good confidence interval estimates to statistically test if individual response functions $f_j(x)$ are non-zero, and thus should be included in the minimal model. Given the marginal posterior $\mathrm{P}(\boldsymbol{f}|\boldsymbol{y}, \boldsymbol{X}, \boldsymbol{\lambda}) = \mathcal{N}(\boldsymbol{f}, \tilde{\boldsymbol{X}} V_\beta \tilde{\boldsymbol{X}}^\top)$, we start from the standard approach using chi-squared statistics $T_r = \hat{\boldsymbol{f}}^\top \boldsymbol{V}_{\boldsymbol{f}}^{r-} \hat{\boldsymbol{f}}$, with $\boldsymbol{V}_{\boldsymbol{f}} = \tilde{\boldsymbol{X}} V_\beta \tilde{\boldsymbol{X}}^\top$, $r$ the rank of the covariance matrix and $\boldsymbol{V}_{\boldsymbol{f}}^{r-}$ its rank-$r$ pseudo-inverse. This statistic is known to up-weight the dimensions of the response space that are most heavily penalized to zero [27], so we further correct the estimate by setting $r$ equal to the effective degrees of freedom of the response function (see Suppl. Info. S5 for final expression).

For the final model selection, the p-values associated with each input feature (obtained from $T_r$ using Farebrothers algorithm for a weighted sum of $\chi^2$ variables) are compared to a pre-defined significance threshold (by default, 0.01); the terms that do not pass this test are removed from the model, followed by an optional refit of the remaining parameters. This model selection procedure completely avoids traditional model comparison, dramatically reducing computation time.

## 5 Numerical results

**Artificial data.** We first confirm the ability of our estimation procedure to recover ground truth covariates in a simple toy example, intended as a minimal version of the naturalistic scenario where task variables are correlated, with only a subset actually driving neural responses. More precisely, we model the spike responses of a single neuron to several correlated inputs. We define 4 relevant input dimensions – two continuous inputs (response functions defined by a RBF basis) and two corresponding to discrete events (convolution kernel given as a difference of two gamma functions). We further introduce a set of nuisance variables, one for each relevant input dimension, which are strongly correlated with the inputs ($r = 0.7$), but do not affect the spike counts. We calibrated the

parameters of the ground truth model to a relatively realistic setting (30min of data, sampled in 6ms bins; 5Hz mean firing rates, see source code for details).

The estimator robustly recovers the ground truth kernels and correctly rejects all nuisance dimensions (Fig. 1B). The good response function estimates also translate into well calibrated p-values for model selection, with our statistical test rejecting all nuisance variables, while preserving all true covariates (Fig. 1C). These confidence bounds are well calibrated and match those produced by much more computationally intensive bootstrapping (see Suppl. Info. S6 and Fig. S1), despite the Gaussian approximation. The model fit remains good even for substantially less data, with small response function m.s.e. errors for as little as 10min of recording time (although the exact limit will depend on the mean firing rate of the cell; Fig. 1D). The model selection performance is good in a similar data range, with type 2 errors very rare once the response function estimates are good (Fig. 1E). Interestingly, the model selection does not seem affected by the fraction of nuisance variables even with only 30min of data (Fig. 1F). Overall, the estimation procedure performs well; the qualitative features of the solution are robust and can be replicated for a wide range of ground truth model parameters.

We directly compared our P-GAM against a standard Poisson GLM with global elastic net ($L_1$ and $L_2$) regularization, known to encourage group sparsity [28]. To keep the comparison as fair as possible, we used the exact same spline basis for both models, which focuses the comparison on regularization and the optimization procedures used for fitting the models to data. For the GLM, the $L_1$ and $L_2$ penalty terms were scaled by a single hyperparameter $\lambda \in \mathbb{R}^+$, optimized using a grid search, with the cross-validated pseudo-$r^2$ as optimization objective (using statsmodels python library [29]). This choice was motivated primarily by the need to keep computation time reasonable. We used artificial data similar to the first set of experiments, but with all covariates being continuous, systematically varying the dimensionality of the input (50% of which are uncorrelated nuisance variables) and the amount of data available, keeping the mean firing rates fixed at 1Hz. We find that the GLM regularization has much less success than our P-GAM in terms of zeroing out responses to nuisance variables, resulting in filters with substantial spurious structure (Fig. 2A). In terms of fit quality, the cross-validated pseudo-$r^2$ was comparable across models, but consistently slightly better for the P-GAM fits (paired Wilcox signed-rank test p<0.001); this is true for a for a wide range of ground truth model parameters (Fig. 2B, C), also seen in terms of the m.s.e. of the response functions (not shown) and for L1-only regularization (see Suppl. Info. Fig. S5). Strikingly, P-GAM achieves these improvements despite requiring much less computing time (Fig. 2D).

One important, but often neglected, statistical feature of experimental data is that the coverage of the input space is not uniform. This is especially true in the naturalistic setting and negatively affects the quality of model estimates, in particular on the tails of the input distribution. To model this scenario, we compare the effects of uniform sampling of the individual input dimensions to alternative more concentrated input distributions (Fig. 2D). Indeed, we find that for both models the overall fit quality decreases the sparser the tails (Fig. 2E). However the GLM is substantially more sensitive to this manipulation, with P-GAM gains becoming substantial for the sparse input distribution (Fig. 2E; Wilcoxon signed-rank test, p<0.001). In particular, the GLM seems to do a much poorer job in estimating the response function at the extremes (Fig. 2F). Note that the exponential nonlinearity means that even relatively small deviations in the response function estimates can translate into large firing rate differences. For instance, if we have a unit with 5 Hz mean firing rate and a kernel gain with peak equal to 2 in log-space, underestimating the kernel by 0.5 would result in a 14.5Hz difference in peak rate, while overestimating by the same amount would yield a 24Hz difference. Overall, P-GAM does better to a degree that matters in practice.

**Macaque PFC recordings.** Given our results using artificial data, P-GAM promises to deliver more robust estimates in experimental data with a lot of correlated input dimensions, nuisance variables and inhomogeneous coverage of the input space. Here, we explore the utility of our model in making sense of prefrontal neural responses in macaques performing a virtual reality spatial navigation task [3]. Briefly, macaques use a joystick with two degrees of freedom to navigate a virtual reality environment; they are trained to find specific targets in this environment, in exchange for a juice reward. Here, task variables include continuous inputs, such as the latent 2D location of the monkey and target within the VR environment; eye position; radial and angular velocities[5]; and discrete events, such as target onset, start and stop of moment, or the timing of reward. A multi-electrode

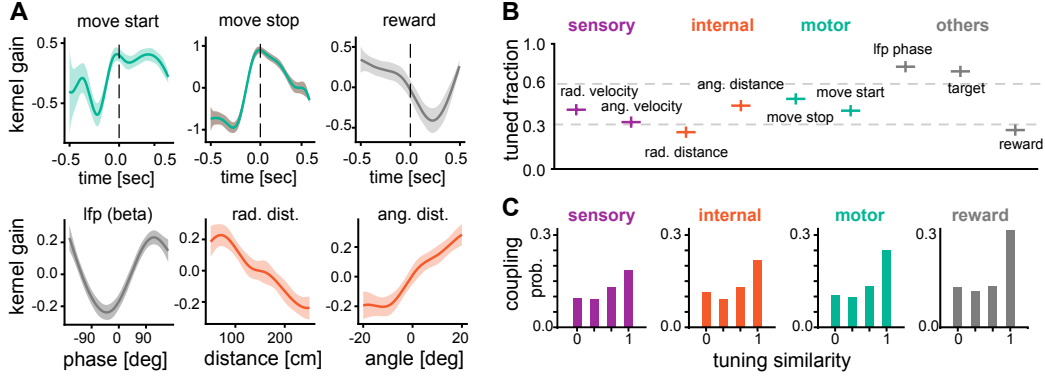

Figure 3: P-GAM fits for monkey PFC recordings during a VR spatial navigation task. **A.** Example of one PFC unit with mixed selectivity: estimated response functions for a subset of 6 task variables. **B.** Fraction of tuned neurons for each individual co-variate. **C.** Coupling probability as a function of tuning similarity for different input variables.

array simultaneously records neural responses in prefrontal cortex. A typical session lasts about 90min, with spike counts measured in 6ms time bins; the analysis presented here includes 30 sessions from one animal.

To investigate the tuning properties of neurons in this task, we fit our coupled P-GAM to neural responses, regressing a full list of possible relevant inputs, which includes all the task features mentioned before plus a range of internal features, in particular couplings across all neurons, and filtered LFP in several bands (15 input dimensions). Putting it all together, this results in a model with over 500 $\beta$ parameters and about 150 $\lambda$ hyperparameters. We find that an overwhelming fraction of the units (90%) exhibit mixed selectivity, with neural responses being driven by several task variables (see example unit in Fig.3A). Given the large amount of data, and the broad mixed selectivity of the neural responses, GLMs with elastic net regularization perform quite well on this dataset as well, but the estimates are less robust as the amount of data used for fitting decreases (see Suppl. Info. S8 and Suppl. Fig. S4). Overall, each individual task dimension ends up encoded in the neural activity to some degree. As expected, local oscillations explain part of neural variability in most cells (Fig.3B). More interestingly, a large fraction of the cells are tuned to the target onset, potentially reflecting the working memory component of the task, since the target is only shown transiently and needs to be kept in memory while trying to reach it.

Motivated by traditional work in early sensory coding that shows a systematic alignment between stimulus and noise correlations [18], we further investigated the structure of the model couplings with respect to the tuning similarity of the the two cells involved (using minimal model, with non-significant terms removed). Here we evaluate similarity as $s = 1 - 0.5||\tilde{f} - \tilde{g}||_2$, where $\tilde{f}$ and $\tilde{g}$ denote a normalized version of the response functions $f$ and $g$ of the two units, for (possibly a subset) of inputs. This metric is 1 (0) when the two cells have identical (orthogonal) tuning. We separately investigate to which degree tuning to different types of input features can explain neural couplings. We find that tuning similarity is indeed correlated with coupling probability, although the effect is stronger for cognitive, relative to sensory, input dimensions (Fig.3C). Qualitatively similar results are obtained when restricting the units to very well separated single units (Suppl. Fig.S6), suggesting that the effect cannot be trivially explained by spike sorting artifacts. This structure is even more remarkable here, since the model explicitly regresses out global circuit dynamics (LFP), which explain a large fraction of neural covariability.

## 6 Discussion

Despite the ubiquity of stimulus-response models for analyzing neural tuning, robustly estimating the parameters of such models remains unexpectedly hard in practice. Here we have shown that casting the problem in the GAM framework allows us to define flexible nonlinear stimulus-response models that can adapt to the complexity of biological responses, including multi-dimensional dependencies. Moreover, we can estimate the parameters of such models efficiently, provide confidence bounds on

the model parameters, and statistically test for the effect of individual inputs on neural responses, to generate interpretable minimal statistical models of neural responses. Our procedure outperforms standard regularized GLMs, in particular in its ability to ignore irrelevant input dimensions. Finally, when applied to neural recordings from a naturalistic foraging task, the model is able to extract several nontrivial features previously demonstrated using traditional approaches that rely on tightly controlled stimuli.

Our solution relies on a large body of work on efficient estimation for low-rank GAMs. The focus here is on bringing individual components together into a tool of practical utility for the experimental setting. Still, we inherit several useful asymptotic guarantees from statistics; in particular, dGCV is known to asymptotically reach optimal MSE reconstruction error, while traditional ML estimators do not [30]. Additionally, as long as the underlying assumptions of the model are correct, it yields smoother estimates and better convergence rates compared to GLM, while doing no worse when the true functions are not smooth [31]. The core ingredient of our particular model is the regularization procedure, which enforces smoothness while shrinking the null space of the constraints to zero. Our approach is similar to ARD [32, 33] in spirit, but for multidimensional parameters; it can also be thought of as something akin to group sparsity, but without the mathematical inconveniences entailed by hierarchical regularization [8]. It is also intuitively closer to our assumptions about the statistical structure of the data compared to simpler alternatives such as elastic nets [28], and computationally less demanding than GP-based alternatives [10, 12, 13], although a new variant of GP-based GAMs [34] may prove competitive for future extensions. Our GAM framework makes model specification intuitive, while the user-friendly library makes it easy to adapt the analysis to new datasets. Lastly, while faster than standard GLM libraries, our P-GAM implementation can still be substantially improved by incorporating new advances in scalable approximate inference, in particular Zoltowski et al [9].

Some would argue that tuning estimation is a relic of a past era when neuroscience technology was restricted to recording one or few neurons at a time. So the field as a whole needs to shift towards population-dynamics-level explanations of circuit computation [35]. This is definitely a fair point; however, we would argue that tuning estimation remains important, and important to be done well. First, from the perspective of an experimental neuroscientist, it is a natural first pass though the data, and a useful tool for exploratory data analysis. Second, from a statistical perspective, understanding the marginal statistics of the conditional neural responses is a critical stepping stone towards better joint models. Moreover, GAMs can serve as building blocks for latent dynamical systems models of population activity [36, 37, 38].

**Broader impact**   We expect that our new fitting procedure and the associated python library will prove of broad utility to scientists in experimental neuroscience looking as a first pass analysis to their data, and to data scientists looking to develop new population level statistical models of neural activity. We do not foresee any potential negative outcomes arising from the availability of such a tool. The nature of this work makes the discussion on biases caused by data, and potential system failures not applicable.

## Acknowledgments and Disclosure of Funding

We thank Stefania Bruni and Eric Avila for their valuable help with neural data collection and preprocessing. We also thank funding sources NRT-HDR 1922658, 1R01MH125571-01, Google faculty research award (CS), U19 NS118246 and R01 DC014678 (DA). The authors declare no competing interests.

## Footnotes

[1] Code available at: https://github.com/BalzaniEdoardo/PGAM.

[2] The same functional form applies for $k(\cdot)$ and $h(\cdot)$.

[3]We drop the neuron index $i$, to simplify notation.

[4]Correspondingly, the firing rate distributions are log-normal.

[5]Note that these variables are naturally strongly correlated.

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
