[Supplementary Material · Suppl_Efficient_estimation_of_neural_tuning_during_naturalistic_behavior.pdf]

# Efficient estimation of neural tuning during naturalistic behavior
# -Supplementary Information-

**Edoardo Balzani**
Center for Neural Science
New York University
New York, NY, 10003
eb162@nyu.edu

**Kaushik Lakshminarasimhan**
Center for Theoretical Neuroscience
Columbia University
New York, NY, 10027
jl5649@columbia.edu

**Dora E. Angelaki**
Center for Neural Science
New York University
New York, NY, 10003
da93@nyu.edu

**Cristina Savin**
Center for Neural Science
Center for Data Science
New York University
New York, NY, 10003
cs5360@nyu.edu

## S1 P-spline basis

We model the individual response functions $f_j, k_j, h_j$ using a penalized spline basis expansion. We use a finite and local basis set of splines (piece-wise polynomials) of order $m$, interpolating between a fixed set of knots. Each $k$-dimensional spline basis of order $m$ with support between $[x_{m+2}, x_{k+1}]$ is specified by selecting $k + m + 2$ interpolation knots $x_1 < x_2 < ... < x_{m+k+2}$. Given the knots, the basis set is defined recursively as [1]:

$$B_i^m(x) = \frac{x - x_i}{x_{i+m+1} - x_i} B_i^{m-1}(x) + \frac{x_{i+m+2} - x}{x_{i+m+2} - x_{i+1}} B_{i+1}^{m-1}(x),$$

such that $i = 1, ..., k$ and $B_i^0 = \begin{cases} 1 & x_i \leq x < x_{i+1} \\ 0 & \text{otherwise} \end{cases}$.

The basis has full support on $[x_{m+2}, x_{i+1})$, where $\sum_i B_i^m(x) = 1$, this interval will be the spline evaluation domain. $x_{m+2}, ..., x_{i+1}$ are called internal knots, while the first and last $m + 1$ knots, which are outside the evaluation domain, are needed only to define the first and initial spline basis element, and can be chosen arbitrarily.

## S2 The effects of penalization: effective degrees of freedom and the smoothing bias

In this section we reparametrize a LS objective so that the estimator parameters have spherical covariance in the absence of penalization [2]; thanks to this parametrization it will be easy to describe how adding a penalty term constrains the model space. This will lead to the definition of the effective degrees of freedom [9] that are used for the statistical testing of covariate significance [4]. For this discussion, $\boldsymbol{X}$ will be the model matrix for a single smooth with penalization parameter $\lambda$ and penalty matrix $\boldsymbol{S}$. Let's suppose that we want to fit the smooth as a penalized least-squares (LS) objective:

$$\|\boldsymbol{y} - \boldsymbol{X\beta}\|^2 - \lambda\boldsymbol{\beta}^\top \boldsymbol{S\beta}.$$

Let's first decompose QR decompose the model matrix $X = QR$, and set $\beta'' = R\beta$, so that the model matrix will be $Q$ and the penalty becomes $R^{-T}SR^{-1}$. We can further apply an eigenvalue decomposition to the penalty:

$$R^{-\top}SR^{-1} = UDU^{\top},$$

and the eigenvalues $D_{ii}$ are arranged in decreasing order. Finally, we can apply a second reparametrization $\beta' = U^{\top}\beta''$, so the corresponding model matrix becomes $QU$, and the penalty reduces to the diagonal $D$. Since $U$ is orthogonal and the columns of $Q$ are orthogonal vectors, under this reparametrization, the unpenalized ML covariace matrix will be:

$$V_{\beta} = (U^{\top}Q^{\top}QU)^{-1}\sigma^2 = I\sigma^2, \tag{1}$$

where $\sigma^2$ is the data variance.
If we apply the penalty, the bayesian covariance matrix becomes:

$$V_{\beta} = (I + \lambda D)^{-1}\sigma^2. \tag{2}$$

The penalized estimator is then

$$\hat{\beta}' = (I + \lambda D)^{-1}U^{\top}Q^{\top}y, \tag{3}$$

and the unpenalized is just $\tilde{\beta}' = U^{\top}Q^{\top}y$.

We can interpret the penalized estimate as a shrinked version of the unpenalized problem,

$$\hat{\beta}_i' = (1 + \lambda D_{ii})^{-1}\tilde{\beta}_i'. \tag{4}$$

The factor $(1 + \lambda D_{ii})^{-1}$ lies in $(0, 1]$, and can be viewed as the *effective* degrees of freedom (EDF) of $\hat{\beta}_i'$. Reversing the reparametrization, the total effective degrees of freedom can then be computed as:

$$\sum_i (1 + \lambda D_{ii})^{-1} = \text{tr}(I + \lambda D)^{-1}$$

$$= \text{tr}\left((QU^{\top}(I + \lambda D)^{-1}UQ^{\top})\right)$$
$$= \text{tr}(Q(I + \lambda UDU^{\top})^{-1}Q^{\top})$$
$$= \text{tr}(Q(R^{-T}R^{\top}Q^{\top}QRR^{-1} + \lambda S)^{-1}Q^{\top})$$
$$= \text{tr}(QR(R^{\top}Q^{\top}QR + \lambda S)^{-1}R^{\top}Q^{\top})$$
$$= \text{tr}(X(X^{\top}X + \lambda S)^{-1}X^{\top})$$
$$= \text{tr}((X^{\top}X + \lambda S)^{-1}X^{\top}X) = \text{tr}(F). \tag{5}$$

If $\lambda \to \infty$, then the effective EDF will be the multiplicity of the zero eigenvalue of the penalty matrix, and the maximum EDF will be the number of parameters when $\lambda = 0$.
Since the unpenalized estimator is unbiased, $\text{E}[\hat{\beta}_i'] = (1 + \lambda D_{ii})^{-1}\beta_i$, so the penalty represents the relative smoothing bias. Reverting again the reparametrization we see that $F$ represents the smoothing bias. Using $\beta' = U^{\top}R\beta$, therefore,

$$\text{E}[\hat{\beta}] = R^{-1}U\text{E}[\hat{\beta}']$$
$$= R^{-1}U(I + \lambda D)^{-1}\beta'$$
$$= R^{-1}(I + \lambda UDU^{\top})^{-1}U\beta'$$
$$= R^{-1}(I + \lambda R^{-T}SR^{-1})^{-1}U\beta'$$
$$= (R^{\top}R + \lambda S)^{-1}R^{T}U\beta'$$
$$= (X^{\top}X + \lambda S)^{-1}R^{T}UU^{\top}R\beta$$
$$= (X^{\top}X + \lambda S)^{-1}X^{\top}X\beta$$
$$= F\beta.$$

In the natural re-parametrization, the degree of penalizaiton of each parameter do not affect others and $D_{ii}$ directly indicate the relative penalization of the i-th model component. Fixed the penalty $S$,

$\boldsymbol{D}$ is uniquely defined, and the penalty's action is to suppress some dimensions of the model space. Which dimensions are effectively more suppressed depends on the penalty matrix, the penalty we are using affects the wiggliness of the smooths, so that increasing $\lambda$ lead to increasingly smoother models.

In general, for a Poisson GAM with penalty matrix $\boldsymbol{S_\lambda}$ the EDF can be computed as:

$$\text{tr}(\boldsymbol{F}) = (\boldsymbol{X}^\top \boldsymbol{W} \boldsymbol{X} + \boldsymbol{S_\lambda})^{-1} \boldsymbol{X}^\top \boldsymbol{W} \boldsymbol{X}, \tag{6}$$

which as the same form of (5), but substituting $\boldsymbol{X}$ with $\sqrt{\boldsymbol{W}}\boldsymbol{X}$ and $\lambda\boldsymbol{S}$ with $\boldsymbol{S_\lambda}$ and $\boldsymbol{W}$ is the weight matrix from the PIRLS algorithm at convergence.

## S3  Gradient and Hessian of the GCV

Here we provide gradient and Hessian of the GCV score, following the derivation proposed in [9]. First of let's set $\boldsymbol{y} = \sqrt{\boldsymbol{W}}\boldsymbol{z}$ and $\boldsymbol{X}' = \sqrt{\boldsymbol{W}}\boldsymbol{X}$, so that,

$$\text{GCV}(\boldsymbol{\lambda}) = \frac{n\|\boldsymbol{y} - \boldsymbol{A}\boldsymbol{y}\|^2}{\{n - \text{tr}(\boldsymbol{A})\}^2},$$

and $\boldsymbol{A} = \boldsymbol{X}'(\boldsymbol{X}'^\top \boldsymbol{X}' + \boldsymbol{S_\lambda})^{-1} \boldsymbol{X}'^\top$. We will drop the primes for what follows.

Let's QR-decompose $\boldsymbol{X} = \boldsymbol{Q}\boldsymbol{R}$, and find a square root of the matrix $\boldsymbol{S_\lambda} = \boldsymbol{B}^\top \boldsymbol{B}$, for example using Cholesky decomposition.
After augmenting the matrix $\boldsymbol{R}$ with $\boldsymbol{B}$, and applying a singular value decomposition we obtain:

$$\begin{bmatrix} \boldsymbol{R} \\ \boldsymbol{B} \end{bmatrix} = \boldsymbol{U}\boldsymbol{D}\boldsymbol{V}^\top.$$

The rank deficiency of the problem is dealt by removing the singular values that are too low, in respect to the maximum singular value (if $D_{ii} < \max(\text{diag}(\boldsymbol{D}) \cdot \sqrt{\varepsilon})$ , with $\varepsilon$ the machine precision, remove column and row $i$ of $\boldsymbol{D}$, and remove columns $i$ of $\boldsymbol{V}$ and $\boldsymbol{U}$).
Defining $\boldsymbol{U}_1$ as the sub-matrix of $\boldsymbol{U}$ such that $\boldsymbol{R} = \boldsymbol{U}_1 \boldsymbol{D}\boldsymbol{V}^\top$, we have

$$\boldsymbol{X} = \boldsymbol{Q}\boldsymbol{U}_1\boldsymbol{D}\boldsymbol{V}^\top$$
$$\boldsymbol{G} = \boldsymbol{X}^\top \boldsymbol{X} + \boldsymbol{S_\lambda} = \boldsymbol{V}\boldsymbol{D}^2\boldsymbol{V}^\top$$
$$\boldsymbol{A} = \boldsymbol{X}\boldsymbol{G}^{-1}\boldsymbol{X}^\top = \boldsymbol{Q}\boldsymbol{U}_1\boldsymbol{U}_1^\top \boldsymbol{Q}^\top.$$

This immediately leads to

$$\text{tr}(\boldsymbol{A}) = \text{tr}(\boldsymbol{U}_1\boldsymbol{U}_1^\top). \tag{7}$$

The only computationally expensive step is the QR decomposition, that has to be performed once, but the rest is cheap. Setting $\rho_j = \log\lambda_j$, we will find the gradient and hessian of the GCV in this transformed parameters in order to force positive weights and handle more easily large penalties. We first note that $\boldsymbol{G}^{-1} = \boldsymbol{V}\boldsymbol{D}^{-2}\boldsymbol{V}^\top$. Using standard multivariate calculus,

$$\frac{\partial \boldsymbol{G}^{-1}}{\partial \rho_j} = -\boldsymbol{G}^{-1}\frac{\partial \boldsymbol{G}}{\partial \rho_j}\boldsymbol{G}^{-1} = -\lambda_j \boldsymbol{V}\boldsymbol{D}^{-2}\boldsymbol{V}^\top \boldsymbol{S}_j \boldsymbol{V}\boldsymbol{D}^{-2}\boldsymbol{V}^\top$$

$$\frac{\partial \boldsymbol{A}}{\partial \rho_j} = \boldsymbol{X}\frac{\partial \boldsymbol{G}^{-1}}{\partial \rho_j}\boldsymbol{X}^\top = -\lambda_j \boldsymbol{Q}\boldsymbol{U}_1\boldsymbol{D}^{-1}\boldsymbol{V}^\top \boldsymbol{S}_j \boldsymbol{V}\boldsymbol{D}^{-1}\boldsymbol{U}_1^\top \boldsymbol{Q}^\top. \tag{8}$$

Using the chain rule, the second derivatives can be obtained as:

$$\frac{\partial^2 \boldsymbol{G}^{-1}}{\partial \rho_j \partial \rho_k} = \boldsymbol{G}^{-1}\frac{\partial \boldsymbol{G}}{\partial \rho_j}\boldsymbol{G}^{-1}\frac{\partial \boldsymbol{G}}{\partial \rho_k}\boldsymbol{G}^{-1} - \boldsymbol{G}^{-1}\frac{\partial^2 \boldsymbol{G}}{\partial \rho_j \partial \rho_k}\boldsymbol{G}^{-1} + \boldsymbol{G}^{-1}\frac{\partial \boldsymbol{G}}{\partial \rho_k}\boldsymbol{G}^{-1}\frac{\partial \boldsymbol{G}}{\partial \rho_j}\boldsymbol{G}^{-1}$$

$$\frac{\partial^2 \boldsymbol{G}}{\partial \rho_j \partial \rho_k} = \delta_j^k \lambda_j \boldsymbol{S}_j$$

$$\frac{\partial^2 \boldsymbol{A}}{\partial \rho_j \partial \rho_k} = \boldsymbol{X}\frac{\partial^2 \boldsymbol{G}^{-1}}{\partial \rho_j \partial \rho_k}\boldsymbol{X}^\top.$$

Finally, we get:

$$\frac{\partial^2 \mathbf{A}}{\partial \rho_j \partial \rho_k} = \lambda_j \lambda_k \mathbf{Q} \mathbf{U}_1 \mathbf{D}^{-1} \mathbf{V}^\top \left[ \mathbf{S}_k \mathbf{V} \mathbf{D}^{-2} \mathbf{V}^\top \mathbf{S}_j \right]^\ddagger \mathbf{V} \mathbf{D}^{-1} \mathbf{U}_1^\top \mathbf{Q}^\top + \delta_j^k \frac{\partial \mathbf{A}}{\partial \rho_j}, \tag{9}$$

where $\mathbf{M}^\ddagger = \mathbf{M} + \mathbf{M}^\top$, $\delta_j^k = 1$ if $j = k$ and 0 otherwise.

Let's define $\alpha = \|\mathbf{y} - \mathbf{A}\mathbf{y}\|^2$, and $\delta = n - \mathrm{tr}(\mathbf{A})$, $\mathbf{y}_1 = \mathbf{U}_1 \mathbf{Q}^\top \mathbf{y}$, $\mathbf{M}_j = \mathbf{D}^{-1} \mathbf{V}^\top \mathbf{S}_j \mathbf{V} \mathbf{D}^{-1}$, and $\mathbf{F}_j = \mathbf{M}_j \mathbf{U}_1^\top \mathbf{U}_1$, we have,

$$\frac{\partial \delta}{\partial \rho_j} = \lambda_j \mathrm{tr}(\mathbf{F}_j)$$

$$\frac{\partial^2 \delta}{\partial \rho_j \partial \rho_k} = -2\lambda_j \lambda_k \mathrm{tr}(\mathbf{M}_k \mathbf{F}_j)$$

$$\frac{\partial \alpha}{\partial \rho_j} = \lambda_j \mathbf{y}_1^\top \left( 2\mathbf{M}_j - \mathbf{F}_j - \mathbf{F}_j^\top \right) \mathbf{y}_1$$

$$\begin{aligned}
\frac{\partial^2 \alpha}{\partial \rho_j \partial \rho_k} = &-\lambda_j \lambda_k \mathbf{y}_1^\top \big( 2\mathbf{M}_k \mathbf{M}_j + 2\mathbf{M}_j \mathbf{M}_k - \mathbf{M}_j \mathbf{F}_k - \mathbf{M}_k \mathbf{F}_j \\
&- \mathbf{F}_k^\top \mathbf{M}_j - \mathbf{F}_j^\top \mathbf{M}_k - 2\mathbf{F}_k \mathbf{M}_j \big) \mathbf{y}_1 \\
&+ \delta_j^k \lambda_j \mathbf{y}_1^\top \left( 2\mathbf{M}_j - \mathbf{F}_j^\top - \mathbf{F}_j \right) \mathbf{y}_1, {}^{[1]}
\end{aligned}$$

Finally, we obtain:

$$\mathrm{GCV} = \frac{n\alpha}{\delta^2} \tag{10}$$

$$\frac{\partial\, \mathrm{GCV}}{\partial \rho_j} = \frac{n}{\delta^2} \frac{\partial \alpha}{\partial \rho_j} - \frac{2n\alpha}{\delta^3} \frac{\partial \delta}{\partial \rho_j} \tag{11}$$

$$\frac{\partial^2\, \mathrm{GCV}}{\partial \rho_j \partial \rho_k} = -\frac{2n}{\delta^3} \frac{\partial \delta}{\partial \rho_k} \frac{\partial \alpha}{\partial \rho_j} + \frac{n}{\delta^2} \frac{\partial^2 \alpha}{\partial \rho_j \partial \rho_k} - \frac{2n}{\delta^3} \frac{\partial \alpha}{\partial \rho_k} \frac{\partial \delta}{\partial \rho_j} + \frac{6n\alpha}{\delta^4} \frac{\partial \delta}{\partial \rho_k} \frac{\partial \delta}{\partial \rho_j} - \frac{2n\alpha}{\delta^3} \frac{\partial^2 \delta}{\partial \rho_j \partial \rho_k}. \tag{12}$$

## S4 Confidence interval average coverage probability

Here, we sketch the proof of the average coverage properties of the Bayesian confidence intervals following the argument in [7]. What we want to show is that we can choose some constants $c_i$, $i = 1, ..., n$ and $d$ such as,

$$\frac{1}{n} \sum_i \mathrm{P}(|\hat{f}(x_i) - f(x_i)| \leq c_i d) = 1 - \alpha, \tag{13}$$

for some level $\alpha$, where $\hat{f}(x_i)$ is the estimated smooth component at $x_i$ and $f(x_i)$ is the true value. The same problem can be formulated by defining a variable $I \sim \mathrm{Uniform}(\{1, ..., n\})$ and asking requiring that

$$\mathrm{P}(|\hat{f}(x_I) - f(x_I)| \leq c_I d). \tag{14}$$

The following discussion holds for Gaussian distributed $\mathbf{y}$ with error variance $\sigma^2$ and identity link; the general exponential family case follows exactly the same steps with $X' = \sqrt{\mathbf{W}} X$ in place of $X$ (where $\mathbf{W}$ is the weight matrix from the PIRLS), and the scale parameter in place of $\sigma^2$ [7].

Let's define two random variables $V := \{\hat{f}(x_I) - E[\hat{f}(x_I)]\}/c_I$ and $B := \{E[\hat{f}(x_I)] - f(x_I)\}/c_I$, so that the variable whose distribution we want to find is $V + B$. Due to the results on the estimator distribution, defining $\tilde{\mathbf{X}}$ such as $\hat{\mathbf{f}} = \tilde{\mathbf{X}} \hat{\boldsymbol{\beta}}$, we have that $\hat{\boldsymbol{\beta}}$ is Gaussian (asymptotically Gaussian in the general case), and therefore so is $\hat{f} = \tilde{\mathbf{X}} \hat{\boldsymbol{\beta}}$. Since $I$ is uniformly distributed, we can conclude that $V$ is a Gaussian mixture with 0 mean. If we choose $c_i$ appropriately, so that each Gaussian in the mixture component has the same variance, $V$ will be Gaussian. The confidence intervals are computed under two main assumption:

- the bias $B \approx 0$

- $\mathrm{Var}[B]$ is very small with respect to $\mathrm{Var}[V]$.

Under this assumptions, we can approximate the distribution of $V + B$ as a Gaussian with mean approximately 0. Due to the gaussianity, in order to fully specify the distribution of $V + B$, all we need to do is compute $\mathrm{Var}[V + B]$. Setting $\boldsymbol{C} = \mathrm{diag}(c_i)$ we have,

$$
\begin{aligned}
\mathrm{Var}[V + B] &= \mathrm{E}\left[\sum_i \frac{(\hat{f}(x_i) - f(x_i))^2}{c_i^2} P(I = i)\right] \\
&= \frac{1}{n}\mathrm{E}[\|\boldsymbol{C}^{-1}(\hat{\boldsymbol{f}} - \boldsymbol{f})\|^2] \\
&= \frac{1}{n}\mathrm{E}\left[\|\boldsymbol{C}^{-1}\tilde{\boldsymbol{X}}(\hat{\boldsymbol{\beta}} - \boldsymbol{\beta})\|^2\right],
\end{aligned}
\tag{15}
$$

where the expectation is over the distribution of $\hat{\beta}$.
For any matrix $\boldsymbol{B}$, we have that

$$
\begin{aligned}
\mathrm{E}\left[\|\boldsymbol{B}(\hat{\boldsymbol{\beta}} - \boldsymbol{\beta})\|^2\right] &= \mathrm{E}\left[\|\boldsymbol{B}(\hat{\boldsymbol{\beta}} - \mathrm{E}[\hat{\boldsymbol{\beta}}])\|^2\right] + \|\boldsymbol{B}(\mathrm{E}[\hat{\boldsymbol{\beta}}] - \boldsymbol{\beta})\|^2 \\
&= \mathrm{tr}(BV_{\hat{\boldsymbol{\beta}}}\boldsymbol{B}^\top) + \|\boldsymbol{B}(\boldsymbol{F} - \boldsymbol{I})\boldsymbol{\beta}\|^2 \\
&\simeq \mathrm{tr}(BV_{\hat{\boldsymbol{\beta}}}\boldsymbol{B}^\top) + \mathrm{E}_\pi[\|\boldsymbol{B}(\boldsymbol{F} - \boldsymbol{I})\boldsymbol{\beta}\|^2],
\end{aligned}
\tag{16}
$$

with $\boldsymbol{F} = (\boldsymbol{X}^\top \boldsymbol{X} + \boldsymbol{S}_\lambda)^{-1}\boldsymbol{X}^\top \boldsymbol{X}$ and $V_{\hat{\boldsymbol{\beta}}} = (\boldsymbol{X}^\top \boldsymbol{X} + \boldsymbol{S}_\lambda)^{-1}\boldsymbol{X}^\top \boldsymbol{X}(\boldsymbol{X}^\top \boldsymbol{X} + \boldsymbol{S}_\lambda)^{-1}\sigma^2$ and $\pi \sim \mathcal{N}(0, \boldsymbol{S}_\lambda \sigma^2)$ is the prior over the parameters. Noting that $\boldsymbol{F} - \boldsymbol{I} = -(\boldsymbol{X}^\top \boldsymbol{X} + \boldsymbol{S}_\lambda)^{-1}\boldsymbol{S}_\lambda$, we can conclude that

$$
\begin{aligned}
\mathrm{E}\left[\|\boldsymbol{B}(\hat{\boldsymbol{\beta}} - \boldsymbol{\beta})\|^2\right] &\simeq \mathrm{tr}(BV_{\hat{\boldsymbol{\beta}}}\boldsymbol{B}^\top) + \mathrm{E}_\pi\left[\mathrm{tr}\left\{\boldsymbol{B}(\boldsymbol{F} - \boldsymbol{I})\boldsymbol{\beta}\boldsymbol{\beta}^\top(\boldsymbol{F} - \boldsymbol{I})^\top\boldsymbol{B}^\top\right\}\right] &(17) \\
&= \mathrm{tr}(BV_{\hat{\boldsymbol{\beta}}}\boldsymbol{B}^\top) + \sigma^2\mathrm{tr}\{\boldsymbol{B}(\boldsymbol{F} - \boldsymbol{I})\boldsymbol{S}_\lambda^{-1}(\boldsymbol{F} - \boldsymbol{I})^\top\boldsymbol{B}^\top\} \\
&= \sigma^2\mathrm{tr}\left\{\boldsymbol{B}\left[(\boldsymbol{X}^\top \boldsymbol{X} + \boldsymbol{S}_\lambda)^{-1}\boldsymbol{X}^\top \boldsymbol{X}(\boldsymbol{X}^\top \boldsymbol{X} + \boldsymbol{S}_\lambda)^{-1}\right.\right. \\
&\quad \left.\left. +(\boldsymbol{X}^\top \boldsymbol{X} + \boldsymbol{S}_\lambda)^{-1}\boldsymbol{S}_\lambda(\boldsymbol{X}^\top \boldsymbol{X} + \boldsymbol{S}_\lambda)^{-1}\right]\boldsymbol{B}^\top\right\} \\
&= \sigma^2\mathrm{tr}\left\{\boldsymbol{B}(\boldsymbol{X}^\top \boldsymbol{X} + \boldsymbol{S}_\lambda)^{-1}\boldsymbol{B}^\top\right\} \\
&= \sigma^2\mathrm{tr}\left\{\boldsymbol{B}^\top \boldsymbol{B}(\boldsymbol{X}^\top \boldsymbol{X} + \boldsymbol{S}_\lambda)^{-1}\right\}.
\end{aligned}
\tag{18}
$$

Now, using (18) and setting $\boldsymbol{B} = \boldsymbol{C}^{-1}\tilde{\boldsymbol{X}}$, we obtain,

$$
\mathrm{E}\left[\|\boldsymbol{B}(\hat{\boldsymbol{\beta}} - \boldsymbol{\beta})\|^2\right] \simeq \mathrm{tr}\left\{\tilde{\boldsymbol{X}}^\top \boldsymbol{C}^{-2}\tilde{\boldsymbol{X}}(\boldsymbol{X}^\top \boldsymbol{X} + \boldsymbol{S}_\lambda)^{-1}\right\}\sigma^2/n
\tag{19}
$$

$$
= \mathrm{tr}\left\{\boldsymbol{C}^{-2}\tilde{\boldsymbol{X}}(\boldsymbol{X}^\top \boldsymbol{X} + \boldsymbol{S}_\lambda)^{-1}\tilde{\boldsymbol{X}}^\top\right\}\sigma^2/n = \sigma^2,
\tag{20}
$$

where the last equality holds by setting $\boldsymbol{C} = \mathrm{diag}(\tilde{\boldsymbol{X}}(\boldsymbol{X}^\top \boldsymbol{X} + \boldsymbol{S}_\lambda)^{-1}\tilde{\boldsymbol{X}}^\top)^{1/2}$, this choice also equalizes the variance of the mixture components. Finally, in order to satisfy (14), all we need to do is set $d = -\sigma^2 z_{\alpha/2}$, with $z_{\alpha/2}$ the $\alpha/2$ critical point of standard normal distribution. Since GCV is asymptotically optimal in the MSE sense, the assumptions about the bias variance of estimates should hold when of smoothing penalties are selected by GCV optimization.

## S5  Statistical significance for individual input dimensions.

We want to test if individual response functions $f_j(x)$ are non-zero, and thus should be included in the minimal model. Given the marginal posterior for $\boldsymbol{\beta}$, $\hat{\boldsymbol{f}} = \tilde{\boldsymbol{X}}\hat{\boldsymbol{\beta}} \sim \mathcal{N}(\boldsymbol{f}, \tilde{\boldsymbol{X}}V_\beta\tilde{\boldsymbol{X}}^\top)$, we start from the standard approach using chi-squared statistics $T_r = \hat{\boldsymbol{f}}^\top \boldsymbol{V}_{\boldsymbol{f}}^{r-}\hat{\boldsymbol{f}}$, with $\boldsymbol{V}_{\boldsymbol{f}} = \tilde{\boldsymbol{X}}V_\beta\tilde{\boldsymbol{X}}^\top$, $r$ the rank of the covariance matrix and $\boldsymbol{V}_{\boldsymbol{f}}^{r-}$ its rank-$r$ pseudo-inverse. This statistic is known to up-weight the

dimensions of the response space that are most heavily penalized to zero [8], so we further correct the estimate by setting $r$ equal to the effective degrees of freedom of the response function [8].

The resulting test statistic is: $T_r = \hat{\boldsymbol{f}}^\top \boldsymbol{M}^{r-} \hat{\boldsymbol{f}}$, with

$$\boldsymbol{M}^{r-} = U \begin{bmatrix} \lambda_1^{-1} & & & & \\ & \ddots & & & \\ & & \lambda_{k-2}^{-1} & & \\ & & & B & \\ & & & & 0 \end{bmatrix} U^\top, \ B = \Lambda \tilde{B} \Lambda, \ \Lambda = \begin{bmatrix} \lambda_{k-1}^{-1} & 0 \\ 0 & \lambda_k^{-1} \end{bmatrix}, \ \tilde{B} = \begin{bmatrix} 1 & \rho \\ \rho & \nu \end{bmatrix},$$

$B = \Lambda \tilde{B} \Lambda, \ \Lambda = \begin{bmatrix} \lambda_{k-1}^{-1} & 0 \\ 0 & \lambda_k^{-1} \end{bmatrix}, \ \tilde{B} = \begin{bmatrix} 1 & \rho \\ \rho & \nu \end{bmatrix}$, where $k = \lfloor r \rfloor + 1$, and $\lambda_1 \geq \lambda_2 \cdots \geq \lambda_n > 0$ the sorted non-zero eigenvalues of $\boldsymbol{V}_f$, with $\nu = \tau - k + 1$ and $\rho = \sqrt{\nu(\nu-1)/2}$ and $U$ the matrix of eigenvectors sorted according to the eigenvalues magnitude. Under the hypothesis that $f(x) = 0$, $T_r \sim \chi_{k-2}^2 + \nu_1 \chi_1^2 + \nu_2 \chi_1^2$, with $\nu_1 = [\nu + 1 + \sqrt{1 - \nu^2}]/2$ and $\nu_2 = \nu + 1 - \nu_1$.

## S6 Confidence interval quality

We tested the coverage properties of the approximated confidence intervals by means of numerical simulations. In particular, we simulated a thousand 30 minutes long spike count vectors at 6ms resolution according to the GAM generative model for the toy problem described in the main text. We then fit the GAM to each independent spike count vector, and computed the corresponding empirical 95 % confidence intervals as a bootstrap estimate of the ground truth. Results show that the approximate confidence intervals estimated through our proposed procedure match very well to these numerical estimates, (Fig. S1).

Figure S1: Confidence interval quality. Approximate confidence intervals estimated by our theoretical procedure (red shaded area) overlapped with bootstrap estimates (gray lines).

## S7 Multidimensional filters.

P-spline based GAM can be readily extended to multidimensional response filters such as spatio-temporal responses, or any other multivariate interaction. For notation simplicity we will describe the case of a bivariate response function $f(x, y)$. The approach that we take here is to expand the function using a tensor product of the basis set $\{a_j(x)\}$ and $\{b_j(y)\}$ covering the $x$ and $y$ domain respectively [6]. The expansion takes the form of,

$$f(x, y) \approx \sum_{ij} \beta_{ij} a_i(x) b_j(y).$$

As in the univariate case, we assume that the likelihood of the GAM is that of a Poisson with mean,

$$\log(\mu_t) = \sum_{i=1}^{M} \sum_{j=1}^{N} \beta_{ij} a_i(x_t) b_j(y_t)$$
$$= \boldsymbol{X}_t \boldsymbol{\beta},$$

where $\boldsymbol{X} \in \mathbb{R}^{T \times MN}$ is the model matrix and can be derived by the model matrix of the marginals as $\boldsymbol{X} = \boldsymbol{X}_x \odot \boldsymbol{X}_y$, with $\odot$ being the row-wise Kronecker product, see figure S2.

tensor product basis element

Figure S2: Two-dimensional tensor product basis element.

In order to measure the wiggliness of the response function we could use,

$$J(f) = \lambda_x \int \left( \frac{\partial^2 f}{\partial x^2} \right) dx dy + \lambda_y \int \left( \frac{\partial^2 f}{\partial y^2} \right) dx dy,$$

Following [3], this integral can be approximated as,

$$J(f) \approx \lambda_x \boldsymbol{\beta}^\top \left( S_x \otimes \boldsymbol{I} \right) \boldsymbol{\beta} + \lambda_y \boldsymbol{\beta}^\top \left( \boldsymbol{I} \otimes S_y \right) \boldsymbol{\beta},$$

where $\otimes$ is the Kronecker product and $S_x$ and $S_y$ are the penalty matrices of the marginals (e.g. $S_x = \int \boldsymbol{a}'' \boldsymbol{a}''^\top dx$).

We tested this approach on published data from tetrode recordings from rat CA1 neurons during open field exploration (one example cell from the supplementary material of [5]; data includes spike counts, LFP phase, speed and position recorded at 1250Hz, for a total of 18sec of activity). It is well known that CA1 responses are modulated by both LFP theta phase and the position of the rat on a linear track. Moreover, the interaction between the two variables can have significant effects on neural responses [5]. Fitting a two-dimensional P-GAM on this data, we were able to correctly capture this interaction (Fig. S3). Furthermore, model selection excluded additional contributions of the animal's speed on neural responses (p-value, 0.26), supporting the hypothesis that the speed modulation of hippocampal responses can be explained by the speed modulation of theta alone [5].

More concretely, we fit two GAMs, one fully additive and another including the interaction between phase and position,

$$\log \mu_1 = f(x) + g(\phi) + h(v) \tag{21}$$
$$\log \mu_2 = f(x, \phi) + h(v), \tag{22}$$

where $x$ is the rat position, $\phi$ is the LFP phase and $v$ is the speed.

While the fully additive model clearly fails to capture phase-position interactions, the bivariate GAM correctly reconstructs the tuning function of the unit, obtaining a filter that resembles closely the one in [5] (Fig. S3). The fit results readily summarize some of the findings described in the original paper, such as the presence of joint phase-position tuning and the absence of modulation due to speed, without the need to make detailed parametric assumptions about the response shape.

Figure S3: Multidimensional nonlinear filters. Fits on data from rat CA1 single unit for (A) the fully additive model and, (B) a P-GAM with interactions. (C), Speed modulation of neural responses, not significant.

## S8   Filter comparison: PGAM vs. regularized GLM

We compared the tuning estimates obtained based on the monkey PFC neural responses for our PGAM and the elastic net regularized GLM. We randomly selected a 50 minutes long recording session and fit both models to this data with all task variables as regressors; we did not include the coupling terms, as fitting a coupled model would be too time consuming for the GLM. We found that response functions estimated over the full 50 minutes session look similar between the two models (Fig. S4A, compare black lines in top and bottom rows). Overall, the fit quality, measured as cross-validated pseudo-$R^2$, is comparable between the two models (Fig. S4B). However, when we restrict the data to the first 20 minute, GAM fits are more stable, consistently matching those obtained from the full session (Fig. S4A top, compare red vs. black lines), whereas GLM fits are more noisy (Fig. S4A top, compare blue vs. black lines) and sporadically underfit, see movement stop and target (Fig. S4A bottom). To compare and summarize these difference across cells, we computed a stability index as:

$$\text{stability index} = 2\frac{\|f_{\text{GLM},50} - f_{\text{GLM},20}\|_2 - \|f_{\text{GAM},50} - f_{\text{GAM},20}\|_2}{\|f_{\text{GLM},50} - f_{\text{GLM},20}\|_2 + \|f_{\text{GAM},50} - f_{\text{GAM},20}\|_2}, \tag{23}$$

where $f_{\text{GAM/GLM},20}$ and $f_{\text{GAM/GLM},50}$ are the P-GAM/GLM estimated responses using 20 and 50 minutes of recording respectively; the higher the stability index, the more robust are the tuning functions estimated from the P-GAM relative to the GLM. Indeed, at the level of the population GLM systematically results in more robust estimates with limited data (Fig. S4A).

Figure S4: Filter stability. (A) Example of response filters estimated with GAM (top row) and GLM with elastic net regularization (bottom row) for a full recording session (black lines) or a subset of 20 minutes (colored lines, red P-GAM, blue GLM, yellow shaded area represent time interval of the target presentation). (B) cross-validated pseudo-$R^2$ difference, not statistically significant. (C) Fit stability. Percent variation of the stability index, equation (23), for the P-GAM estimates relative to GLMs.

Figure S5: Comparison of cross-validated pseudo-$r^2$ values for P-GAM relative to a GLM with L1 regularization. Equivalent to Fig.2B, but for the L1-regularized GLM.

Figure S6: Coupling probability as a function of tuning similarity for well-isolated single units. Coupling probability and tuning similarity are estimated as described in the main text.

## Footnotes

[1] Modified from the original reference [9].