[Reviews · NeurIPS 2020]

Review 1

Summary and Contributions: [Post author reply:] The authors have satisfied my comments, and I increase their score to a 7. I would recommend authors: - discuss previous work with GPs, and make it clear the advantages of your method over a GP prior - discuss how your approach compares with a group-penalized GLM when no continuous inputs are considered (i.e., only z_j(t) which need not be binary). Readers will want to quickly know which situations are best for your approach---make this clearer. This approach is not for mapping spatio-temporal receptive fields. Also make clear that your approach is not applicable for settings in which the desire is to estimate a temporal filter with an estimated nonlinear link function (unless I'm missing something, your approach cannot do that). - include results (could be in the supplemental) about estimating a standard GLM for the real data in Fig 3) - also make clear if your method fails when the input covariates are correlated, as may be the case in natural behavior. How do correlated inputs affect the results, especially for estimates of nonlinear link functions? ----- This paper proposes going beyond GLMs to use generalized additive models (Poisson-GAMs) to characterize the relationship between a neuron's activity and its external input, input from other neurons, and its prior activity history. An important point is that P-GAM can identify *nonlinear* transformations between external input (e.g., behavior, stimuli, etc.) and neural activity. This is akin to identifying "tuning" curves, which are often nonlinear, for neurons. The model also can provide certainty estimates for each covariate---helpful for neuroscientists that must decide which features are most important for a neuron to encode. The authors verify their method via simulations and apply P-GAM to neural activity recorded from the prefrontal cortex of macaque monkeys. The authors also provide some interesting ways of optimizing and regularizing their model, which is of interest to modelers.

Strengths: The claims and technical contributions appear sound. I think the work is significant, given that GLMs are used all over the place in neuroscience. I also imagine the parameter estimation, unique penalties, and parameter certainty estimation will be of interest to the NeurIPS community and modelers in general. The figures are also clear, minimal, and quite beautiful. I am giving this paper a 6 due to some weaknesses (below), but I can be convinced to give it a 7 after seeing the rebuttal and other reviews.

Weaknesses: I have two main weaknesses for the work. 1. Allowing for nonlinear filters for continuous inputs x(t) is nice, but changes in x(t) over time likely are important for predicting spikes, which is ignored by this model. In addition, choosing to operate on each dimension separately seems it will not perform well for a high number of dimensions. For example, GLMs are typically used to estimate receptive fields of neurons, where the input is an image of hundreds of pixels. For P-GAM, each pixel would be given its own nonlinear filter, and the model would not directly smoothly vary its weights for nearby pixels. Is this true? How would P-GAM hold up for image inputs? Or are you suggesting P-GAM (used for low-d abstract variables, like angle/velocity) is complementary to GLMs? If you could add a linear filter inside the GAM (i.e., a dot product between beta and x(t), before passing beta^T * x(t) through f), this would allow both for high-d inputs as well as time. Right now, the applicability for the method is small. However, perhaps the authors could run a simulation/test with their real data what happens when including past history as input variables (e.g., distance for the past 10 time points)---and assess if the filters make sense over time. The second weakness is that the authors do not consider a Gaussian process prior or compare to work that uses a GP to map inputs to neural activity (papers given in "prior work" section). I imagine a spectrum, from GLM to P-GAM to a GP prior, and there must be trade-offs. The authors should compare their results to already proposed methods with GP priors, and at the very least discuss the tradeoffs between GAM and GP-like methods. - why not compare to a GP prior? - no linear product for x(t) inputs? so no time?

Correctness: To my knowledge, the claims and technical soundness is correct. One comment--- Fig. 3C, it doesn't seem there is a relationship except for neurons with very high tuning similarities (last bar is high, all other bars are about the same height). You should check to make sure this is not simply due to increased firing rates (e.g., neurons with higher firing rates are more likely to have higher coupling prob. and higher tuning similarity).

Clarity: Overall, the paper was well-written and clear, and the figures were amazing. Here are a few comments to help improve the paper for readers: - Fig. 1A, it appears the "nonlinear filters" label applies to all filters, but it's really just the first two (the ones for continuous inputs). Make this clearer. - Fig. 1E, unclear what is plotted here. Misses/false positives of predicting a spike or not? - For OCV and dGCV, please provide the reader intuition for the numerator and denominator in (7) and above equation. Why is it Ay? Why does large alpha lead to smoothness? A few sentences will help the reader (instead of staring at the equations for a while). - Somewhat unclear at first that no convolution/dot product is taken for x(t) (continuous input). Thus, any changes in x(t) cannot be taken into account. Most people using GLMs are expecting to consider time with their variables. - lines 138-152. I would reword here. I'm not entirely sure why you want S_\lambda to be full rank. Are you worried that S_\lambda with low rank will have too much freedom to overfit? Right now it seems large penalties for S_\lambda will disfavor nonsmooth functions already---so already large large penalties favor null responses. Clearly I'm confused here! - line 248-249 (last sentence), follow up on the implications of this---right now, reader may not know why this is remarkable. - Fig. 3A, give more intuition for what the nonlinear filter shapes mean. E.g., what does a linear ramp for angle imply? Why does reward drop. This would help to show off the interpretability of your method.

Relation to Prior Work: I think a weakness of the paper is not comparing this method to methods that utilize a GP prior. Perhaps the authors know something that I don't (i.e., maybe these methods are painfully slow), but they should still cite and make it clear in their paper. The most similar one is "Active learning of neural response functions with Gaussian processes" by Park et al., NIPS 2011. Similar to this is "Bayesian active learning of neural firing rate maps with transformed gaussian process priors" by Park et al., Neur Comp 2014. I would also at the least cite "Gaussian process based nonlinear latent structure discovery in multivariate spike train data" by Wu et al., NIPS 2017. There, they put a GP on the tuning curves (although they have a latent variable model). Please discuss how your work relates to these prior studies.

Reproducibility: Yes

Additional Feedback: A few comments, no need to reply to them: - Please put the link to source code in either the intro (as a footnote) or in the Discussion or Broader impact---somewhere a reader can find it easily vs. on page 6. - line 94, remind reader why dGCV is important (why are we considering it here?) - avoid saying "P-GAM ... model" like in Fig 1. The M in GAM stands for model already. - line 148, do you mean (6) instead of (2)? (2) has no objective function. - line 189, make clear you are considering a Poisson GLM (I'm assuming you are doing this.) - line 266, what is GML? - broader impact: I would soften the language of "not applicable"---just remove the last sentence. Who knows if a nefarious person can make it applicable in some way.


Review 2

Summary and Contributions: In this paper, the authors develop a Poisson distribution-based GLM to fit the spike trains of individual neurons as a function of a potentially wide variety of behavioral and internal neural variables. They consider aspects of regularization and cross validation, and show results on both artificial datasets and a database of macaque monkey PFC neurons.

Strengths: The paper is generally employing a collection of existing tried and tested methods, but to excellent effect. The results on artificial data are pretty good, showing the power of the method - although the improvement in the fit over a elastic net regularized GLM is rather modest, and the extent of the test was not so great. The authors are admirably precise about all the steps they are taking and the contribution (largely the specific way that regularization and optimization works), making the paper a pleasure to read.

Weaknesses: The paper is describing and providing a method; but I was ultimately a bit disappointed about the application to the macaque data. It really didn't teach us a great deal (and it wasn't clear that we learned a lot more than we would have from the regularized GLM method in this case). The one notional result is that there is a high coupling probability as a function of multi-dimensional tuning similarity. It would have been very interesting to have investigated this in more detail - certainly, as an expression of functional connectivity, it has some interpretative issues. The paper is refreshingly frank in the discussion at declaring itself to be a fantastic modern method for a problem (single cell encoding model construction/tuning function estimation) that is no longer so popular. I agree with the paper that estimation remains important - but, again, a more direct look at this through the medium of some actual data would have been very interesting. More mechanistically, in the comparison with regularized GLMs, the authors point out that their major differences (if you fix the spline for tuning functions) are optimization and regularization. I would have liked some more information about the optimization aspects.

Correctness: Yes. I didn't check the code - but the descriptions in the paper seem fine.

Clarity: yes - the paper is very well written and clear.

Relation to Prior Work: The relationship with existing work is described rather briefly. It seemed a bit unfair to criticise other methods for being 'nontrivial to regularize well', when a lot of the current paper concerns aspects of regularization too. And the empirical test that was done was against a generic method rather than from one of the cited schemes from the Pillow and/or Paninski labs that have gone to similar amounts of trouble (and applied them to more compelling natural datasets).

Reproducibility: Yes

Additional Feedback:


Review 3

Summary and Contributions: This paper tackles the problem of tuning estimation in a potentially large-scale neural recording. In particular, the goal of the proposed method is to determine a minimal subset of input features to which each of the measured neurons is tuned; they achieve this by using a smoothing prior that enforces sparseness. Their method finds a set of relevant input features for each individual neuron in the measured population. To model the neurons' spiking activity, the method uses a Poisson GAM (Generalized Additive Model); one of the results of the paper is that the GAM achieves the goal better than the more conventional GLM.

Strengths: This paper presents a well-constructed set of methods on a solid ground of previous works. It uses good practices of statistical modeling and testing, both for parameter estimation and model selection. If developed and distributed in a user-friendly package, the method has a potential to become a part of routine analyses in experimental laboratories that perform large-scale neural recording, providing an intuitive "first pass through the data" as the authors point out. To the NeurIPS community, this work demonstrates a successful combination of existing works, mostly from machine learning and statistics, into a useful context of neural data analysis.

Weaknesses: A twist of the last strength element (that this work is a useful combination of existing works) means that this work is less strong on the novelty scale; although this does not make the method less valuable. Given that the greatest significance of this work might be in being a practical analysis tool in the experimental labs, and given the many components in the method, it is not clear how straightforward it would be to actually apply this method to a new dataset. For example, I see possible difficulties in hyperparameter selection, or the identification of all known features and how to appropriately incorporate them into the model. I hope the authors will be able to comment on these issues.

Correctness: The method appears to be correct, although this method incorporates many different existing works and I do not claim a full understanding of these previous works. The method was tested in a reasonable way, using artificial data that demonstrates its ability to select the relevant features against nuisance variables.

Clarity: The paper is well written overall, with clear descriptions of the model. Given that the method makes a rather heavy use of multiple existing methods, I think the authors did a good job balancing the level of exposition in the main text, such that the reader can appreciate the main idea for each part of the method. Minor points: - Possible typo in Eq. 6: argmax? - The parameter alpha in Eq. 8 may be confused with alpha in Eq. 7.

Relation to Prior Work: The paper acknowledges that "the focus of this work is to incorporate "individual components together into a tool of practical utility for the experimental setting", and the corresponding previous works are clearly referenced. Also, in the results section, the method (based on Poisson GAM) is explicitly compared to the previous, GLM-based models.

Reproducibility: Yes

Additional Feedback: - How is the proposed approach related to the method of ARD (Automatic Relevance Determination) prior? - I don't see how the inline expression of S_{f_j} in L63 can be obtained from the second derivative of f(x), between Eqs. 3 and 4. Could you clarify? - It would have been convenient if all equations were labeled. ======= ** Edit ** The author response addressed my question about the applicability of the method, and clarified its relationship to the ARD prior (which was helpful). I remain enthusiastic about this paper.


Review 4

Summary and Contributions: This paper introduces a new variant Poisson GLM model, called P-GAM, that incorporates new regularization techniques and automatic regularization parameter tuning (via a form of generalized cross validation). The authors describe this method and its derivation in detail, then test it on both simulated data, where it is compared to an existing GLM model, and on real neural recordings from a non-human primate. The primary advantages of the P-GAM model are its efficiency (especially for doing model/feature selection) and somewhat improved prediction performance over existing models.

Strengths: The P-GAM method seems highly elegant and a clever solution to the exploding computational demands of model selection in high-dimensional settings. It combines techniques from many different sources: PIRLS for optimization, dGCV for selecting regularization parameters, regularization that pushes model parameters toward the null space of the penalty matrix, etc. Further, despite being a large and complex beast, the model and its derivation are very well and clearly explained.

Weaknesses: 1. The weaknesses of this paper stem only from the model evaluation, which seems to have been given somewhat short shrift. In particular, one element that is missing is the comparison between P-GAM and the earlier GLM on real neural data. These models are compared on simulated data (Figure 2), where P-GAM successfully outperforms the GLM. Yet the analysis on real data (Figure 3) only shows results from the P-GAM model. While these results look fine and believable, without the explicit comparison it is not clear that any different conclusions would be drawn from a standard GLM model applied to the same dataset. While the computational efficiency of P-GAM seems like a good argument for using it over the GLM, without a comparison of (1) prediction performance or (2) interpretability on real neural data, I'm left unconvinced that this offers a real advance. If the models perform similarly but the GLM requires much more computation (e.g. to do model selection), then comparing models with a fixed computational budget would clearly show the advantage of P-GAM. 2. I'm somewhat mistrustful of confidence intervals inferred from analytic posterior distributions on parameters. These posterior CIs can be highly influenced by distributional assumptions in the model that may only have a modest effect on prediction performance. Thus in order to trust the CIs provided by the model I would prefer to see validation against non-parametric CI estimation, obtained for example by bootstrapping the training data.

Correctness: No concerns.

Clarity: The paper was very clear and well written.

Relation to Prior Work: No concerns.

Reproducibility: Yes

Additional Feedback:

[Author Response · NeurIPS 2020]

We thank the reviewers for their thoughtful and detailed comments. Brief replies follow below.

**R1) Input dimensionality:** P-GAMs have similar scaling with the number of input dimensions as traditional GLMs. Nonetheless, using P-GAMs makes most sense when the inputs are task or cognitive variables; we don't envision our methods being commonly used for estimating V1 RFs. This does not mean that P-GAMs are not generally useful: many of the new datasets in systems neuroscience (higher cortices, hippocampus) naturally lend themselves to our analysis.

**Spatio-temporal filters:** We chose to include temporal filters only for binary events, which helps interpretability in the context of our task. P-GAMs can also model spatio-temporal effects: we simply need to add one extra temporal filter for each input dimension, somewhat similar in flavor to the factorization in [Park & Pillow, 2013]. Alternatively, we can define 2D nonlinear spatio-temporal filters for each input dimension, each contributing one additive term to the final GAM expression. Both versions scale linearly with the number of inputs (with a potentially large hidden constant in the second version). We will include examples of this functionality in the new version of the paper (results and code demo).

**Smoothness priors:** our current regularizer already encourages smoothness. Incorporating general GP priors seems more difficult. GP regression is known to scale unfavorably with the number of inputs, requiring additional structure (e.g. Kronecker) to keep computation tractable [Wilson et al, 2014]; even with the extra tricks, GP-based tuning estimates are restricted to low dimensional inputs (less than 10, inapplicable to our data, see e.g. [Savin & Tkacik 2016]). A new variant of GP-based GAMs may prove competitive [Adam, Durrande & John, 2018] but, to our knowledge, this has not yet been adapted to neural tuning estimation. We will cover the GP literature in the discussion.

**Fig1E:** shows misses and false positives for detecting whether an input dimension drives neural responses or not.

**Fig3C:** We've done additional statistics to disentangle the role of mean firing on tuning and coupling strength. A partial correlation analysis confirmed that coupling patterns cannot be trivially explained by differences in mean firing rates.

**R2) Neural implications:** We could not elaborate on monkey data results due to space limits; journal paper to follow.

**GLM comparison:** We chose to compare P-GAMs against a vanilla 'Pillow GLM' instead of a fancier version because this is what most neuroscientists end up using in practice; the group sparsity regularization would be closest in aims, but, to our knowledge, that has not been extensively applied to real data. We are happy to expand on the links to these alternative methods in the introduction/discussion.

**R3) Usability, adapting the code to new datasets:** Up to now, we have used the estimator on 3 different datasets (the monkey one, rat hippocampus and OFC). The initial P-GAM model specification takes a bit of work on any new dataset, but there are relatively few knobs that have to be set by hand (the spline basis); once the model is set, the code runs smoothly — a rotation student with limited coding background managed to get everything done in a few weeks.

**ARD:** The ARD regularizer corresponds to a factorized Gaussian prior with $\beta$-specific variances (treated as hyperparameters). Assuming a 1-to-1 map between parameters and inputs, the prior for irrelevant input dimensions will end up sharply concentrated around zero. It is not clear how to do this in the nonlinear case, when several $\beta$s are used to model each input dimension (one per basis vector). We need some form of group sparsity and traditional ARD can't do that.

**R4) GLM vs P-GAM on real data:** we can include GLM filter estimates and additional P-GAM vs. GLM fit quality quantification on real data in the supplementary info. Full GLM model comparison is intractable for this dataset so we can only do elastic net regularization. In brief, everything is much messier, although some of the trends persist.

**Validation of CIs:** Fair enough. We can definitely get bootstrapping-based CIs for artificial data, probably also for real units although it may prove too computationally expensive to do extensively. We will add those in the updated version. Apart from that, please note that the numerical estimates for the type 1 and 2 errors made by our significance test (Fig3C) are sensible – even if, admittedly, we could only do the validation for artificial data.

**Utility and relevance:** In practice, there are many experimental scenarios where simple GLMs don't quite suffice; most experimentalists will find the technical details too intimidating to attempt complex hierarchical regularization; they also often don't have the computational resources to do brute force model comparison for large models. We are providing a straightforward and relatively general way to get the job done, one that stays tractable even for large datasets.

[Meta-Review · NeurIPS 2020]

This paper nicely presents a methodological contribution for the analysis of neural data acquired during naturalistic settings. The problem considered is relevant as a basic analysis tool for practitioners and the technical contributions are meaningful. This work will primarily be of interest to the subset of the community focused on statistical techniques for neuroscience data. All reviewers viewed the paper favorably.